# Spatial and Temporal Changes of Sand Mining in the Yangtze River Basin since the Establishment of the Three Gorges Dam

**DOI:** 10.3390/ijerph192416712

**Published:** 2022-12-13

**Authors:** Yugai Ma, Yingying Chai, Y. Jun Xu, Zijun Li, Shuwei Zheng

**Affiliations:** 1Department of Resource and Environment, Shijiazhuang University, Shijiazhuang 050035, China; 2College of Geography and Environment, Shandong Normal University, Jinan 250358, China; 3Coastal Studies Inst, Louisiana State University, Baton Rouge, LA 70803, USA; 4Dongying Institute, Shandong Normal University, Dongying 257092, China

**Keywords:** sand mining, spatial and temporal changes, governance strategy, Three Gorges Dam, Yangtze River Basin

## Abstract

The global demand for sand and gravel is at 50 billion tons per year, far exceeding global resource capacities. It reached 7.6 billion tons in 2021 in the Yangtze River Basin (YRB), China. However, production is severely limited in the YRB. Therefore, the incongruity between the supply and demand of river sand is prominent. Wise management of decreasing sand resources in the YRB has become critical since the Three Gorges Dam became operational in 2003. This study synthesized spatial and temporal changes in sand mining activities and quantities along the Yangtze River and its major tributaries from 2004 to 2020. Results from the study show that the mining amount during the period reached 76.2 million tons annually. At the same time, riverine suspended sediment discharge (SSD) downstream of the Three Gorges Dam decreased largely. SSD reduction leads to riverbed erosion, further limiting the riverine sand and gravel sources for mining. Thus, alternative sand and gravel resources, as well as optimizing supply/demand balance, are necessary for sustainable development. There is an urgent need to assess the relationship between river sand resources and exploitation in the YRB for creating a sand and gravel data management system in order to cope with the increasing incongruity between their supply and demand.

## 1. Introduction

Over the last 20 years, the global demand for sand and gravel has tripled. This demand has been projected to double by 2060, driven by the construction and expansion of urban infrastructure [1,2]. While many coastal areas in the world have been losing land, sand and gravel sources have become crucial for protecting and/or creating land in the cities and towns near the coast. For example, 517 million tons (MT) of sand and gravel have been used in land reclamation over the past two decades in Singapore; most of which was imported [3,4]. Therefore, sand and gravel reserves face multiple challenges worldwide.

As high-quality resources, river sand and gravel are highly beneficial to humans. On the other hand, sand mining activities often have profound negative impacts on the ecological environment of rivers [5]. These negative impacts [6,7,8] include changing the river channel morphology and affecting the stability of the river regime, leading to an increase in flood frequency and intensity and exacerbation in drought occurrences [9,10,11,12]. For example, the Manimala River suffers from groundwater level recession and water resource shortages due to river sand mining in southwestern India [13]. Moreover, massive sand mining in rivers could accelerate beach erosion in their deltaic areas [14,15,16]. Illegal sand mining activities are typically random and ignore environmental impacts, causing more harm to the physical and ecological functioning of river systems. Unfortunately, these activities continue worldwide, especially in Malaysia, India, Cambodia, and South Africa [17,18,19].

In China, sand and gravel demand was approximately 18 billion tons in 2021 (https://www.cssglw.com/; accessed on 3 March 2022). The Yangtze River Basin (YRB) is a crucial source and consumption area for high-quality sand. Sand mining along the YRB began in the late 1970s, and illegal sand mining was rampant owing to “profiteering.” Mining along the Yangtze River (YR) mainstream was banned during the early 21st century, and illegal sand mining vessels moved to Poyang Lake, which is connected with the YR. Studies have reported that these sand mining activities have caused the water level of Poyang Lake to drop in the dry season [20], the subaqueous topography to change [21], and the fish and bird habitats to deteriorate [22]. Hence, China has recently established strict regulations for sand mining management and regulation in the YRB. Moreover, the government authorities have increased the inspection and supervision of sensitive waters, including unannounced inspections of key river sections [23].

However, as China has been rapidly developing, the demand for sand and gravel exceeds the supply. The demand-supply discrepancy makes it impossible to prohibit sand mining (including illegal sand mining). Therefore, scientifically and rationally exploiting and managing river sand resources is critical. This study focuses on spatial and temporal changes in sand mining activities in the YRB, aiming to (1) estimate the amount of historical sand mining, (2) analyze characteristics of spatial and temporal mining changes, and (3) propose recommendations and countermeasures for sand mining in the YRB, which can be applicable to other alluvial rivers in the world.

## 2. Study Area and Methods

The Yangtze River originates in the Qinghai-Tibet Plateau in west China and flows 6300 km eastward before entering the East China Sea (Figure 1) [24]. The YRB serves approximately 40% of China’s population and economy. The Yangtze Delta generates almost 25% of China’s total economic production, although it encompasses below 4% of the country’s land area (https://www.thepaper.cn/; accessed on 4 January 2021). This enormous economic burden requires large amounts of sand and gravel.

During the 1980s, the demand for river sand was low and priced at RMB 2–3 per ton. However, it increased to RMB 15 per ton in 2010. Additionally, the demand for sand and gravel increased to 7.6 billion tons within the YRB during 2021 (https://www.cssglw.com/; accessed on 3 March 2022). In response, the price of river sand rose to RMB 142 per ton in 2021 (http://www.zgss.org.cn/; accessed on 1 March 2022).

Sand mining data were obtained from the Yangtze River Sediment Bulletin (2004–2020; http://www.cjw.gov.cn/; accessed on 30 July 2021) and Jiangxi Provincial Water Resources Department (2004–2016; http://slt.jiangxi.gov.cn/; accessed on 16 June 2016). Other sand mining data collected for the main tributaries and lakes are listed in Table 1. Illegal mining data were derived from previous studies and cases of judicial punishment (Table 2).

Table 3 exhibits the average annual runoff and sediment discharge data from Zhutuo, Cuntan, Yichang, Shashi, Hankou, and Datong, along YR.

The average annual runoff and sediment discharge data for Poyang Lake is shown in Table 4.

Estimation of the total sand mining for the YRB can be as given below:(1)$$S_{total}=S_{1}+S_{2}+S_{3}+S_{4}$$
where *S_total_* is the total amount of sand mined within the YRB from 2004 to 2020, *S*_1_ is the sand mined in the YR mainstream (Figure 2), and *S*_2_ is the amount of sand mined in Poyang Lake during the same period, including estimated sand mined both illegally and legally during 2004–2010 [25] and sand mining planning amount in Poyang Lake during 2011 and 2020, *S*_3_ is the amount of sand mining planned in Dongting Lake and the other tributaries during 2012–2020 (Table 5), and *S*_4_ is the extra illegal mining data reported by judicial bodies for the period of 2013–2020 (Table 2).

## 3. Results

### 3.1. Sand Mining along the YR Mainstem

From 2004–2020, 756.3 MT of sand was mined from the mainstream YR, averaging 44.5 MT per year (standard deviation is 23.5 MT). There was large interannual fluctuation (Figure 2), with a peak in 2018 (102 MT) but a sharp decline in 2020 (21.1MT). Overall, mining in the YR mainstem increased largely in 2008, i.e., more than double the annual amount in the previous years.

Before 2010, sand mining planning was spatially concentrated in the Jiangsu and Hubei Provinces (the middle reach of the YR) (Figure 3). After 2015, mining expanded to Chongqing City (the upper reach of the YR) and Shanghai City (the lower reach of the YR. Currently, sand mining planning is mainly concentrated in Chongqing, Hubei, Jiangxi, Anhui, Jiangsu, and Shanghai. The mined sand is transported to other areas through the “golden waterway” of the YR.

### 3.2. Sand Mining in Poyang Lake

From 2004-2020, 539.6 MT of sand were mined from Poyang Lake at an annual average of 31.7 MT (Figure 4). Overall, there was a downward trend with pronounced fluctuations (Figure 5). For example, the maximum mined amount was 49.9 MT in 2007. Mining was then prohibited in 2008 (Figure 5) and resumed in 2009. However, only 4.8 MT were mined in 2018.

### 3.3. Sand Mining from Tributary Rivers and Reservoirs

Over the past 10 years, sand mining from the major tributary rivers and reservoirs of the YR totaled 837.8 MT (Table 5). These include, among others, Han, Jialing, Wu, Fu, Qu, Qing, Han, Xiang, Zi, Yuan, Li River, and Dongting Lake. This mining quantity within a much shorter period of time was higher than that mined from the YR mainstem as well as from Poyang Lake.

### 3.4. Characteristics of Sand Uses

Based on the utilization, sand mining in this review can be divided into three categories: (1) engineering sand mining (used for land formation), (2) construction and operational sand mining (used for building), and (3) river and waterway dredging sand (purpose is to dredge river channel; can be used for land formation and building) (http://slt.hubei.gov.cn/; accessed on 2 December 2019).

The largest amount of sand (584.0 MT) was obtained through engineered sand mining (Table 6). This accounts for 77.22% of the total amount (Table 6). The maximum amount mined was 79.0 MT in 2013 (Figure 6). However, the minimum amount was 3.7 MT in 2020, which is substantially lower than the annual amounts from 2004–2020 (34.4 MT; Figure 6).

The amount from construction and operational sand mining was 28.5 MT, accounting for 3.77% of the total (Table 6). The largest amount mined was during 2015 (8.5 MT), while the annual average was 1.7 MT (Figure 6).

The amount of river and waterway dredged sand was 143.8 MT, accounting for 19.01% of the total (Table 6). The maximum amount (89.0 MT) was mined in 2018, while the annual average was 8.5 MT (Figure 6).

## 4. Discussion

### 4.1. Estimation of the Actual Sand Mining Amount in the YR

Quantifying the sand mining amount from Poyang Lake is crucial for accurately estimating the total sand mined from the YR. Because illegal sand mining along the YR mainstream was prohibited in the early 21st century, mining activities vessels have moved to Poyang Lake, resulting in excessive illegal mining in the lake over an extended period. Jiang et al. [25] found that the amount of sand mining in Poyang Lake was approximately 2154 MT from 2001–2010, which is 7–8 times the approved amount. According to the ship-port affairs department statistics, Chen [26] stated that the actual amount of sand mining in Poyang Lake was approximately 230–290 MT during 2005–2007, which is 4–5 times the planned amount. Therefore, in this study, we referenced the above studies, and the total amount of sand mining was estimated at 3470 MT in the YRB during 2004–2020, yielding an average annual amount of 204.1 MT.

### 4.2. Discrepancy between Supply and Demand of Sand Resources in the YR

Development of the YR Economic Belt resulted in reserve land resources driving the demand for sand and gravel. The reclamation project expanded from reinforcing embankments, filling ponds and foundations, and farmland improvement to coastal protection, island reclamation, and even national defense construction. For example, dredged sand from the Huangpu River was used to reclaim Fuxing Island in Shanghai. Therefore, sand and gravel resources in the YRB have been in short supply.

Additionally, studies show that the middle and lower reaches of YR, including delta regions, will likely continue to erode in the future [27]. However, the sand discharge in the YRB continuously decreased (Figure 7). Sediment discharge has been limited, particularly since the Three Gorges Dam (TGD) became operational in 2003 [28,29]. During 2004–2018, the TGD trapped 77% of riverine sediment inflow, largely reducing sediment load downstream [30]. For example, the sediment discharge fluctuated from 64 MT in 2004 to 46.8 MT in 2020, with an annual value of 31.3 MT at Yichang Station (Figure 7), far below the annual sediment discharge of 501.0 MT during 1950–2000. Hankou and Datong stations, located in the middle and lower reaches of the YR, demonstrated similar decreases (Figure 7). Therefore, there is a prominent incongruity between the sand and gravel supply and the demand in the YRB.

### 4.3. Coping Strategies and Enlightenment

Based on the comprehensive review of sand mining in the Yangtze River Basin, we propose the following strategies that address the discrepancy between sand and gravel supply and demand. These include

1.Gain more insight into the YR channel evolution and adjust sand mining areas scientifically and dynamically.

The YR has experienced inconsistent channel scouring and deposition over time. Therefore, more studies on the river channel dynamics need to be conducted to identify spatiotemporal erosion-deposition equilibrium. The recoverable area should be dynamically adjusted according to runoff and sediment discharge characteristics. For example, the river section from Yichang to Chenglingji is currently in a state of erosion. Therefore, sand mining should be prohibited along this reach.

2.Reduce demand for natural sand.

Reducing demand for natural sand can be achieved through two approaches. First, mining amount and demand are often not equal on a yearly basis. Therefore, a modeling approach to optimize supply/demand balancing can be used, such as the model developed by Zhai et al. [31]. For instance, if the quantity of sand mining this year exceeds the demand, mining can be adjusted accordingly next year.

Second, research should focus on the use of recycled and alternative materials, such as “green concrete” with lower carbon emissions, ultra-high-performance, fly ash, lightweight, and geopolymer concretes, as well as recycled construction materials [32,33,34,35,36,37]. Alternatively, manufactured sand can replace natural sand. Manufactured sand with a particle size below 4.75 mm is formed by removing soil from rocks, pebbles, and mine tailings and crushing them to size.

3.Strengthen the geological survey of sand and gravel resources in the YR and its tributaries and establish a sand and gravel data management system.

Geological surveys should include location and reserve information [38]. These surveys can play an essential role in sand mining planning, alleviating discrepancies between supply and demand, tracing sand mining sources, and reducing the negative environmental and social impacts.

4.Establish scientific standards for sustainable sand mining.

Application of the quality management method from the plan-do-check-act cycle can aid sand and gravel mining management. Thus, the total amount of sand mining, recoverable area and period, and mining elevation can be planned and monitored. First, the on-site supervision should be strengthened following the plan implementation. Topography, actual sediment supply, and flow velocity variation should then be evaluated to gradually establish technical standards and institutionalize the best practices for sustainable sand mining.

Additionally, natural sand consumption must be reduced during construction [39,40]. Relevant departments can review and verify technical standards for construction projects to avoid an engineering “overdesign” [41,42]. We should reduce the construction of non-essential infrastructure and expand the use of green infrastructures, such as permeable pavements [43,44,45,46].

The middle and lower reaches of the YR are typical alluvial rivers with good-quality river sand, being an important sand resource in China. The recent decadal rapid development of the economy and building industry in the country has led to urbanization along the river. Conflicts between supply and demand have become and will remain challenging. Scientific assessment of the sand and gravel resources and planning reasonable sand mining behavior are prerequisites for maintaining the sustainable development of the YR sand. Establishing a scientific management system and strengthening management and control of sand mining processes are important in the sustainable development of sand resources. Additionally, research on renewable and alternative materials is important in solving the incongruity between the supply and demand of sand in China’s Yangtze River Basin. Furthermore, the assessment and review conducted in this study can also serve as a reference for the mining and management of river sand for other rivers in the world, especially those where illegal sand mining is widespread, such as the Indus, Narmada, Nile, and Congo Rivers.

## 5. Conclusions

Currently, the global demand for sand and gravel is approximately 50 billion tons per year. With this gigantic demand, sand and gravel resources in countries worldwide are facing multiple challenges. Taking the YRB as an example, this study analyzed spatial and temporal changes in sand mining activities. The main conclusions are as follows.

After 2015, sand mining areas in the YR expanded to their upper and lower reaches (Chongqing City and Shanghai City, respectively). Currently, sand mining is mainly distributed in the reaches of Chongqing, Hubei, Jiangxi, Anhui, Jiangsu, and Shanghai.There were 756.3 MT and 539.6 MT of sand mined in the YR mainstream and Poyang Lake during 2004–2020, respectively. The demand for sand and gravel increased to 7.6 billion tons within the YRB during 2021. There is a prominent incongruity between the sand and gravel supply and the demand in the YRB.Based on the current state of sand mining in the YR, reducing natural sand consumption for city construction is necessary. Optimizing supply/demand balance and replacing natural sand with recycled (including recycled construction materials) and alternative materials are essential. Furthermore, the recoverable area should be dynamically adjusted according to the YR evolution. Moreover, a geological survey of sand and gravel resources should be strengthened, and a sand and gravel data management system established.

## Figures and Tables

**Figure 1 ijerph-19-16712-f001:**
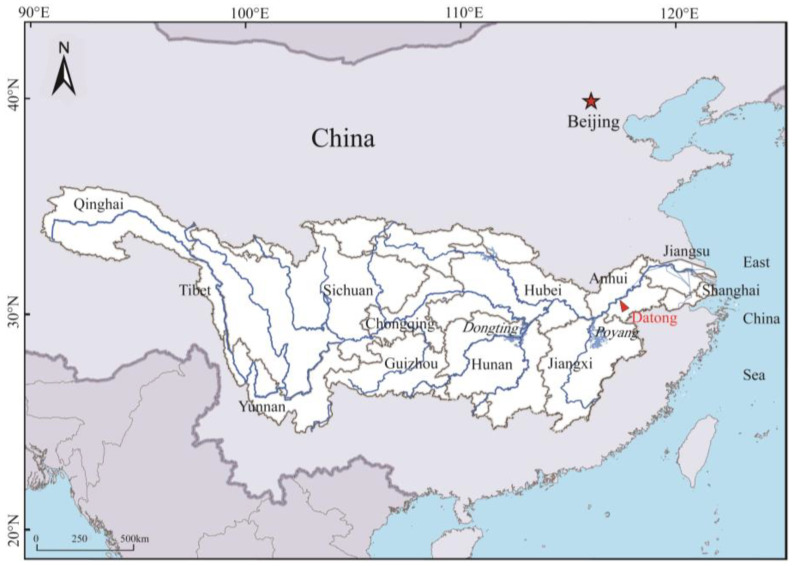
The Yangtze River, its major tributaries, and drainage basin.

**Figure 2 ijerph-19-16712-f002:**
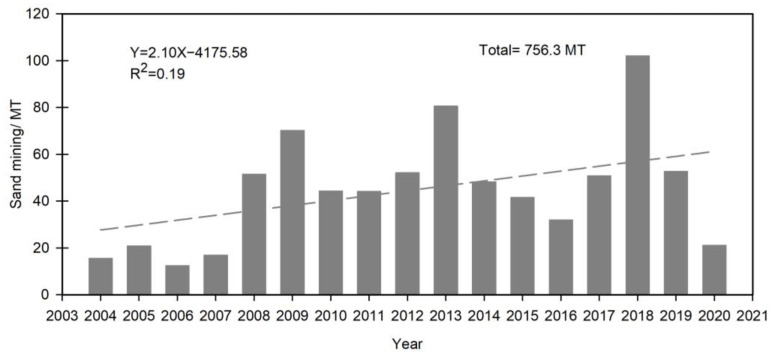
Sand mining planning data for the mainstream YR during 2004–2020.

**Figure 3 ijerph-19-16712-f003:**
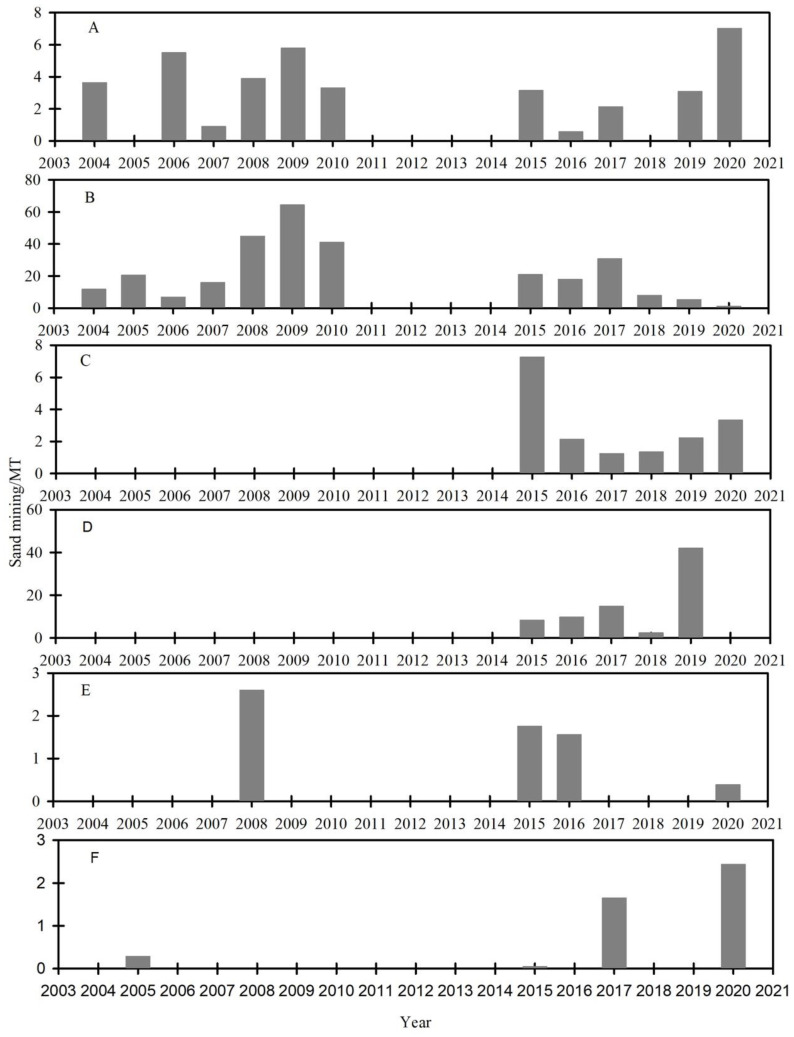
Sand mining within the YRB from 2004–2020: (**A**) Hubei Province; (**B**) Jiangsu Province; (**C**) Chongqing City; (**D**) Shanghai City; (**E**) Anhui Province; (**F**) Jiangxi Province.

**Figure 4 ijerph-19-16712-f004:**
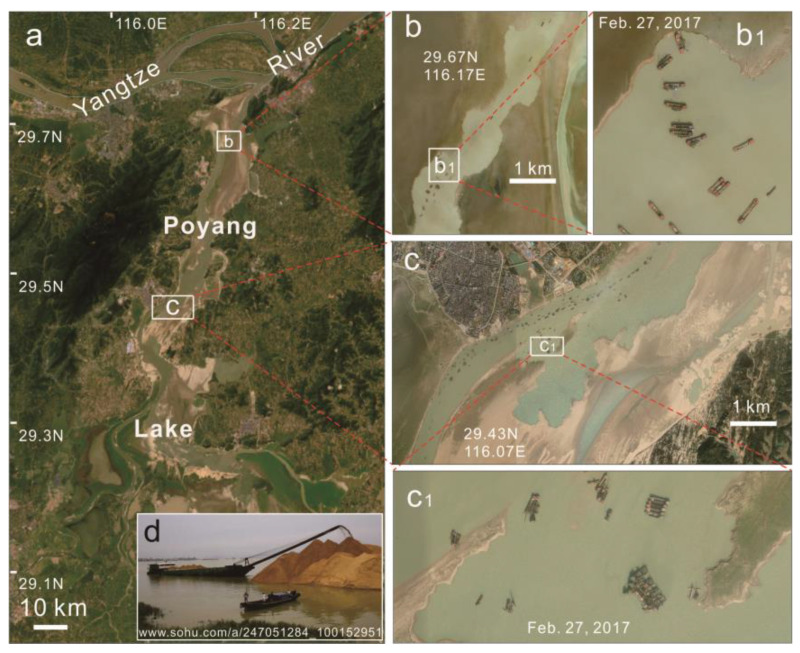
Maps show the location of Poyang Lake. (**a**) Poyang Lake from remote sensing image. (**b**,**c**) Sawtooth sandbar in the Poyang Lake formed by sand mining in 2017. (**b_1_**,**c_1_**) Sand mining equipment in operation on 27 February 2017. (**d**) Photo of a sand dredger in Poyang Lake.

**Figure 5 ijerph-19-16712-f005:**
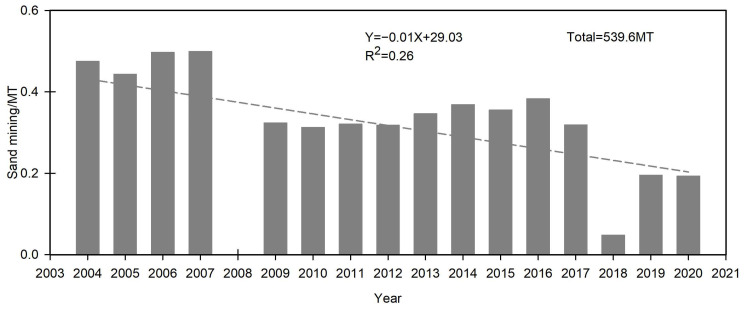
Annual sand mining quantity from Poyang Lake during 2004–2020.

**Figure 6 ijerph-19-16712-f006:**
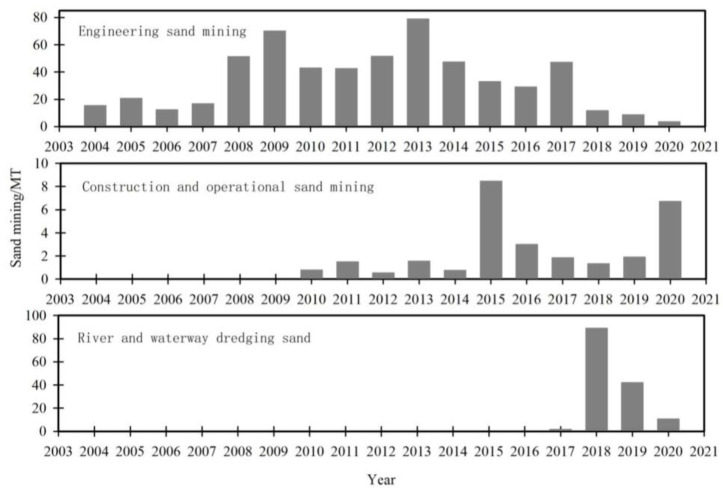
Sand mining by type during 2004–2020.

**Figure 7 ijerph-19-16712-f007:**
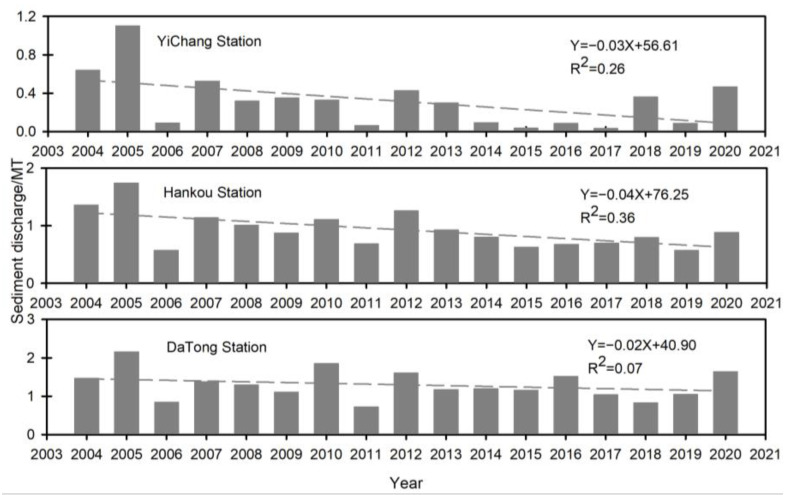
Sediment discharged data from hydrological stations in the mainstream YR from 2004–2020.

**Table 1 ijerph-19-16712-t001:** The planned data source of sand mining in the tributaries of the Yangtze River (YR).

The Main Tributaries	Data Source
Tuo River	Sand Mining Planning Report of Neijiang Section of the Tuo River (2018–2022)
Jialing, Wu, Fu, Qu, Qing River, and other rivers in Chongqing	Sand Mining Planning of Important River Channels in Chongqing (2016–2020)
Han and Dongjing River	Sand Mining Planning for the Middle and Lower Reaches of the Han and Dongjing River (2018–2023)
Xiang, Zi, Yuan, Li River, and Dongting Lake	Sand Mining Planning in the Mainstream of Xiang, Zi, Yuan, Li River, and Dongting Lake in Hunan Province (2012–2016)
	Sand Mining Planning in the Mainstream of Xiang, Zi, Yuan, Li River, and Dongting Lake in Hunan Province (2019–2022)
Gan, Fu, Rao, Xin, and Xiu River	Sand Mining Planning in the Middle and Lower Mainstream of Gan, Fu, Rao, Xin, Xiu River, and Poyang Lake in Jiangxi Province (2019–2023)
	Sand Mining Planning of Poyang Lake in Jiangxi Province (2019–2023)

**Table 2 ijerph-19-16712-t002:** Illegal sand mining in YR.

No.	Time	Location	Sand Mining Amount/×10^3^ Tons	Data Source
1	2001–2010	Poyang Lake	2154 × 10^3^	Literature [25]
2	2005–2007	Poyang Lake	230~290 × 10^3^	Literature [26]
3	2013	Waters of Hannan section, Hubei province	30	https://news.sina.com.cn/; accessed on 18 April 2013
4	2015	Waters of Fuchi Town, Xisai mountain, and Qichun County, Hubei province	881.5	https://hb.ifeng.com/; accessed on 4 June 2019
5	2015–2017	Waters of Zhangjiagang section, Jiangsu province	1000	https://cssglw.com/; accessed on 23 December 2019
6	2016	Waters of Jiangjin section, Chongqing city	170	https://www.sohu.com/; accessed on 1 December 2017
7	2016–2017	Dongting Lake	41 × 10^3^	https://www.thepaper.cn/; accessed on 19 August 2018
8	2016–2018	Waters of Taicang section, Jiangsu province	11 × 10^3^	https://cssglw.com/; accessed on 1 December 2019
9	2017	Waters of Zhenjiang section, Jiangsu province	600	https://www.thepaper.cn/; accessed on 8 August 2018
10	2017	Waters in the forbidden mining area	6	https://www.sohu.com/; accessed on 14 April 2014
11	2015–2018	Waters of Zhangjiagang section, Jiangsu province	12 × 10^3^	https://www.toutiao.com/; accessed on 13 April 2020
12	2017–2020	Waters of the Sichuan province section	46.1	https://www.sohu.com/; accessed on 11 March 2020
13	2018	Waters of Yichang section, Hubei province	14	https://view.inews.qq.com/; accessed on 10 August 2018
14	2018–2019	Waters of Jing River, Hunan province, and Chongqing city, etc.	15 × 10^3^	https://legal.people.com.cn/legal.people.com.cn/; accessed on 31 July 2019
15	2018–2019	Nantong section, Jiangsu province	57.2	http://rgfy.cpcrugao.cn/; accessed on 27 April 2020
16	2018–2019	Dongting Lake	100	https://hnrb.voc.com.cn/; accessed on 15 September 2020
17	2019	Poyang Lake	50	https://www.163.com/; accessed on 7 June 2019
18	2019	Waters in the forbidden mining area	19 × 10^3^	http://www.gov.cn/; accessed on 16 November 2019
19	2019	Waters of Maanshan, Wuhu and Chizhou City, Anhui province and waters of Huangshi City, Hubei province	237	http://www.gov.cn/; accessed on 23 October 2019
20	2019	Waters of Tongling section, Anhui province	5.6	https://k.sina.com.cn/; accessed on 26 May 2020
21	2019	National nature reserve for freshwater dolphins in Tongling, Anhui province (no mining zone)	180	https://ah.ifeng.com/; accessed on 15 June 2020
22	2019	Waters of Zhenjiang section, Jiangsu province and Maanshan section, Anhui province.	51	http://szb.wxrb.com/; accessed on 9 July 2020
23	2020	Waters of Jiangsu and Anhui province sections.	1.0 × 10^3^	http://egal.people.com.cn/; accessed on 6 July 2020
24	2020	Waters of Anhui province	2.5	https://www.chinacourt.org/; accessed on 8 June 2021
25	2020	The waters at the junction of Rugao and Jingjiang City, Jiangsu province.	66.6	https://k.sina.com.cn/; accessed on 23 July 2021

**Table 3 ijerph-19-16712-t003:** Average runoff and sediment discharge in YR.

Year	Hydrological Station	Zhutuo	Cuntan	Yichang	Shashi	Hankou	Datong
2004–2020	annual runoff /billion m^3^	261.4	336.7	395.5	390.6	690.3	875.4
2004–2020	annual sediment dicharge/MT	111.1	136.9	31.3	47.1	92.7	129.8

**Table 4 ijerph-19-16712-t004:** Average runoff and sediment discharge in Poyang Lake.

Year	Tributary	Gan River	Fu River	Xin River	Rao River	Xiu River	Hukou Waterway
	Hydrological Station	Wai zhou	Lijiadu	Meigang	Hushan	Wanjiabu	Hukou
2004–2020	annual runoff/billion m^3^	68.8	12.1	18.3	6.9	3.6	151.6
2004–2020	annual sediment discharge/MT	2.4	1.1	1.1	1.1	0.2	10.0

**Table 5 ijerph-19-16712-t005:** Planned data of sand mining in the tributaries of the YR.

Main Tributaries	Time	Annual Planned Sand Mining Amount/MT	Total Sand Mining/MT
Tuo River	2018–2020	0.9	2.7
Jialing, Wu, Fu, Qu, Qing River, and other rivers in Chongqing	2016–2020	6.5	32.5
Han and Dongjing River	2018–2020	7.0	21
Xiang, Zi, Yuan, Li River, and Dongting Lake	2012–2016	120	600
	2019–2020	59.5	119
Gan, Fu, Rao, Xin, and Xiu River	2019–2020	31.3	62.6
Total planned sand mining amount			837.8

**Table 6 ijerph-19-16712-t006:** Total sand mining data of different user types in the mainstream of the YR.

Type of Sand Mining	Engineering Sand Mining	Construction and Operational Sand Mining	River and Waterway Dredging Sand
Sand mining amount /MT	584.0	28.5	143.8
proportion	77.22%	3.77%	19.01%

## Data Availability

Not applicable.

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
