# Peer review of "Spatial and Temporal Changes of Sand Mining in the Yangtze River Basin since the Establishment of the Three Gorges Dam"

_ijerph, 2022, doi:10.3390/ijerph192416712_

Round 1
Reviewer 1 Report
The paper presents a serious issue strongly affecting the balance of natural systems from river courses to deltaic fringes. The paper is concise, well written and the conclusions are sound.
My only concern is regarding Figure 1 which shows China surrounded by waters, including in the south where it should border other countries.
Author Response
The paper presents a serious issue strongly affecting the balance of natural systems from river courses to deltaic fringes. The paper is concise, well written and the conclusions are sound.
[Authors’ response]: Thank you for your support and for taking time to review our manuscript.
My only concern is regarding Figure 1 which shows China surrounded by waters, including in the south where it should border other countries.
[Authors’ response]: Thank you for catching this mistake. The figure has been modified.
Reviewer 2 Report
Article is devoted to the interesting problem. The sand and gravel demand is big and there is a necessity of estimation of this influence on the environment. It is particularly interesting to assess the impact of mining on the quality of water bodies.
Work has classical structure. Introduction is devoted to the analyse of the different studies . Methods should be improved, this part doesn't give clear information about research.
The cited references are mostly recent publications (within the last 5 years) and relevant, more than 1/2 were published after 2015.
Author Response
Article is devoted to the interesting problem. The sand and gravel demand is big and there is a necessity of estimation of this influence on the environment. It is particularly interesting to assess the impact of mining on the quality of water bodies.
[Authors’ response]: We would like to sincerely thank you for taking your valuable time to review our manuscript. All your comments and suggestions have been taken into account while revising our manuscript, and they have been very helpful for improving the quality of our manuscript.
Work has classical structure. Introduction is devoted to the analyze of the different studies. Methods should be improved; this part doesn't give clear information about research.
[Authors’ response]: Thank you for your comments. In the revised manuscript, we have done our best to improve the section by adding more information and rewriting text to enhance readability.
The cited references are mostly recent publications (within the last 5 years) and relevant, more than 1/2 were published after 2015.
Thank you for your careful reading. Your time is very much appreciated.
Reviewer 3 Report
General comment
An interesting and useful analysis of sand mining in the Yangtze River in China. The paper brings together a wealth of data of the extraction of sand and aggregate from different sections of the river including its lakes and determines a large gap between the supply and the demand for materials. The paper provides a number of worthwhile proposals to help manage the resource on a more sustainable basis.
Specific comments
Line 12 “The sand and gravel demand is at 50 billion tons per year” – this needs to refer to it being the global demand is 50 BT/year.
Line 70 Is this price for 2021?
Line 81 Table 3 Are these figures the average for that period? It should state Average annual runoff/ discharge. Ditto for Table 4.
Line 90 What is the standard deviation for the period?
Line 100 Can you explain why there are many years without any sand extraction in most of these provinces? Surely the demand for materials continued through all the years.
Line 104 Poyong Lake. It would be of interest to readers to include a photograph of the lake.
Has any research been conducted of the impact of sand extraction in these rivers and lakes on aquatic life - fish, dophins, etc and on bird life? Surely it has a significant impact.
Line 119-120 Can you explain these categories? What is engineered sand mining and what is it used for? Similarly for the other two.
Line 133 Table 6 Are these figures the total materials extracted for the entire period 2004-2020?
Line 138 – 144 Surely this indicates a failure of regulation and monitoring of illegal mining. “excessive illegal mining over an extended period” – how could this be allowed to persist?
Line 146 My understanding of your formula is that you add:
?1 is the sand mined in the YR mainstream (2004–2020)
?2 is the amount of sand mining planned in the tributaries and Dongting Lake (2012–2020)
?3 is the amount of sand mined in Poyang Lake (2004–2020)
?4 is the illegal mining data reported by judicial bodies (2013– 2020)
This appears to be double counting. S3 and 4 will be reporting the same location so the amounts will be double counted. This needs further explanation. The figure for 2004–2020 of 3470 MT (line 153) is suspect.
Line 165 What year was the Three Gorges Dam made operational?
Line 167 Does the 501 MT figure a total for the period or an annual figure?
Line 170 “incongruity between the sand and gravel capacity and the demand in YRB” I think you mean between the sand and gravel supply and the demand in YRB. Also in line 234.
Line 184 You mention recycled and alternative materials. You should also consider the recycling of construction materials. When a building is demolished to make way for a new one, are its materials sent to landfill? They can be crushed and broken down and used again in construction. There are companies that recycle construction materials including for road fill and other purposes. This would reduce the pressure on other sources of sand and aggregates.
Line 213 “considerable interest has driven the illegal mining phenomenon” I don’t know what you mean by this. How can interest drive illegality?
Line 236 add “and recycled construction materials…”
